# Preparation and Thermal Evaluation of Novel Polyimide Protective Coatings for Quartz Capillary Chromatographic Columns Operated over 320 °C for High-Temperature Gas Chromatography Analysis

**DOI:** 10.3390/polym11060946

**Published:** 2019-06-01

**Authors:** Meng-ge Huangfu, Yan Zhang, Xin-ling Zhang, Jin-gang Liu, Ying-cong Liu, Yi-dan Guo, Qing-yuan Huang, Xiu-min Zhang

**Affiliations:** 1Beijing Key Laboratory of Materials Utilization of Nonmetallic Minerals and Solid Wastes, National Laboratory of Mineral Materials, School of Materials Science and Technology, China University of Geosciences, Beijing 100083, China; 2103180029@cugb.edu.cn (M.-g.H.); 2103170021@cugb.edu.cn (Y.Z.); z92871168@163.com (X.-l.Z.); zhmsdong@163.com (Y.-c.L.); guoyidan8624@163.com (Y.-d.G.); creator2333@gmail.com (Q.-y.H.); 2School of Electrical Engineering, Beijing Jiaotong University, Beijing 100044, China

**Keywords:** polyimides, chromatographic columns, coatings, copolymerization, thermal properties

## Abstract

Protection of intrinsically brittle quartz chromatographic columns (CCs) from breakage or property deterioration in gas chromatography (GC) analysis has become an important research topic regarding high-temperature GC techniques. Polyimide (PI) has proved to be the most suitable protective coating for quartz CCs. In the current research, a series of novel high-temperature-resistant PI coatings for quartz CCs operated over 320 °C have been successfully prepared. For this purpose, the aromatic diamine with a rigid skeleton structure 2-(4-aminophenyl)-5-aminobenzimidazole (APBI) was copolymerized with two aromatic dianhydrides—3,3’,4,4’-benzophenotetracarboxylic acid dianhydride (BTDA) and 4,4’-oxydiphthalic anhydride (ODPA)—and an aromatic diamine with flexible ether linkages—4,4’-oxydianiline (ODA)—by a two-step polymerization procedure via soluble poly(amic acid) (PAA) precursors, followed by thermal imidization at elevated temperatures. The developed PI coatings exhibited good comprehensive properties, including glass transition temperatures (*T*_g_) as high as 346.9 °C, measured by dynamic mechanical analysis (DMA), and coefficients of linear thermal expansion (CTEs) as low as 24.6 × 10^−6^/K in the range of 50–300 °C. In addition, the PI coatings exhibited good adhesion to the fused quartz capillary columns. No cracking, delamination, warpage, or other failures occurred during the 100-cycle thermal shock test in the range of 25–320 °C.

## 1. Introduction

Gas chromatography (GC) analytical techniques have been widely used in modern analysis for more than half a century [1]. Since the 1980s, with the ever-increasing demands of the petroleum industry, environmental chemistry, and catalyst chemistry, high-temperature GC analysis operated at elevated temperatures over 300 °C has been highly desired [2,3]. As one of the most important components of the GC apparatus, capillary chromatographic columns (CCs) are widely used nowadays because of their high separation efficiency. CCs usually consist of a quartz capillary that is coated with a polymer layer [4,5,6,7]. Quartz is a very pure form of silica and is usually characterized by its high purity, high thermal stability, and excellent corrosion-resistant features. The unique characteristics of the quartz capillary have made it the preferred material for GC analysis. However, a drawback of the quartz capillary is its inherently brittle nature; thus, it needs to be protected by flexible polymer coatings in practical applications [8]. Polymers serving as the protective coating for quartz CCs should meet severe property requirements, including excellent thermal and dimensional stability at elevated temperatures, so as to maintain good adhesion with the inner quartz CCs during thermal shock; good mechanical flexibility and toughness to resist winding; high surface hardness to protect the quartz surface from abrasive damage; and good environmental stability.

Polyimide (PI) represents a class of high-performance polymers characterized by excellent combined properties, including high thermal stability, good mechanical and electrical properties, and superior chemical inertness [9,10,11,12,13], thus meeting most of the requirements of high-performance coatings for the protection of quartz CCs. Due to the extraordinary properties of PIs, they are being paid ever-increasing attention in GC analytical fields, such as the stationary phase in packed-capillary gas chromatography [14], even for use as the CCs themselves instead of conventional glass for gas chromatography [15,16] and as the protective coatings for quartz CCs [17]. For applications as protective coatings for quartz CCs, the key properties of PIs, including high-temperature stability, adhesion ability, and mechanical toughness, have to be compromised in practical applications. For example, highly conjugated and rigid substituents are usually introduced into PI coatings in order to enhance the thermal stability. However, the flexibility and toughness of the derived PI coatings are usually simultaneously sacrificed. When PI-coated quartz CCs are wound on a cage with a small radius of curvature, delamination, cracking, or other adhesive failures might occur in the PI coatings. Moreover, improving the thermal stability of PI coatings usually decreases their adhesion strength to quartz CCs. It has been well established that common PI coatings usually exhibit poor adhesion to quartz (silica) substrates due to the absence of active species on the surface of the PI coatings [18,19]. For optimal adhesion to silica, oxides, and most metals, adhesion enhancements of PI coatings must be performed in practical applications. The methodologies for improving the adhesion of PI coatings with silica substrates usually include roughening the surface of the coating by treatment with alkaline or plasma [20], use of an adhesion promoter [21], or introduction of specific substituents with high adhesion strength to the substrate, such as carbonyl groups, ether linkages, hydroxyl substituents [22], and siloxane segments [23]. For example, in the microelectronics industry, PI passivation or protective coatings are spin-coated onto a silicon wafer with adhesion promoters. Some PIs have built-in adhesion promoters (e.g., siloxane chain segments), while others require the application of a separate adhesion promoter or coupler prior to PI application. However, most of the methods mentioned above are not suitable for the development of PI coatings for the protection of quartz CCs, mainly considering the heat stability and process issues. Thus, PI coatings with high thermal stability, good flexibility and toughness, and high adhesion to quartz CCs are highly desired. However, to the best of our knowledge, few works have been concerned with this topic in the literature up to now. Recently, a PI coating with flexible ether linkages and bulky fluorene units was reported for quartz optical fiber protection [24]. Quartz optical fibers have a composition similar to quartz CCs. The ether bonds endowed the PI coating good flexibility and adhesion to the substrate, while the pendant fluorene substituent brought good thermal stability to the PI coating. However, the developed PI coating could only withstand a maximum servicing temperature of 250 °C. PI coatings derived from the poly(amic acid) (PAA) precursors prepared by pyromellitic dianhydride (PMDA) and 4,4′-oxydianiline (ODA) for silica optical fiber protection have also been reported [25]. However, the derived PI coatings had a coefficient of thermal expansion (CTE) value of 40 × 10^−6^/K, which was much higher than that of the quartz optical fibers (0.57 × 10^−6^/K) [26].

In the current work, as one of our continuing efforts to develop high-performance PI coatings for high-tech applications, a series of PI coatings containing flexible carbonyl groups, ether linkages, and rigid-rod benzimidazole units were developed. Benzimidazole units have been proven to possess good thermal stability along with excellent adhesion to various substrates due to their ability to form hydrogen bonds via the highly active imidazole ring [27,28,29,30,31,32,33]. Introduction of the rigid-rod benzimidazole and flexible carbonyl groups might endow the PI coatings with good heat resistance and good adhesion to the surface of quartz CCs. Further, the flexible ether bonds might guarantee good flexibility and toughness of the PI coatings. The effects of the various substituents on the thermal and mechanical properties of the PI coatings were evaluated in detail.

## 2. Materials and Methods

### 2.1. Materials

4,4’-Oxydiphthalic anhydride (ODPA) and 3,3’,4,4’-benzophenone tetracarboxylic acid dianhydride (BTDA) were purchased from Tokyo Chemical Industry Co., Ltd., Tokyo, Japan and dried in vacuo at 180 °C for 12 h prior to use. ODA was purchased from Wakayama Seika Kogyo Co., Ltd., Wakayama, Japan and used as received. 2-(4-Aminophenyl)-5-amino-benzimidazole (4-APBI) was purchased from Changzhou Sunlight Pharmaceutical Co., Ltd., Jiangsu, China and used as received. *N,N*-dimethylacetamide (DMAc) was purchased from Beijing Yili Fine Chemicals, Beijing, China, distilled prior to use, and stored under a 4 Å molecular sieve. The other commercially available reagents were used without further purification. 

### 2.2. Preparation of Polyimide Coatings and CC Fabrication

The PI coatings were prepared by the copolymerization of BTDA, ODPA, ODA, and APBI via a two-step high-temperature polycondensation procedure. For example, PI-5 was prepared as follows: Into a 500 mL three-necked flask equipped with a mechanical stirrer, a cold-water bath (<10 °C), and a nitrogen inlet, newly vacuo-dried BTDA (16.1115 g, 0.05 mol) and ODPA (15.5105 g, 0.05 mmol) powders were added to a stirred solution of ODA (8.0096 g, 0.04 mol) and 4-APBI (13.4556 g, 0.06 mol) in the distilled DMAc (268.7 g) with a solid content of 16.5% by weight. Then, the cold-water bath was removed and the reaction mixture was stirred in nitrogen at room temperature for 24 h to afford a viscous brown solution. The obtained PAA solution (PAA-5) was diluted with DMAc to a solid content of 10–12 wt % monitored by the absolute viscosities of the PAA-5 varnish controlled in the range of 6000–8000 mPa s. The obtained PAA-5 varnish was filtered through a 0.50 μm polytetrafluoroethylene (PTFE) syringe filter to eliminate any particulates that might affect the quality of the cured coating. The purified PAA-5 solution was cast onto a clean borosilicate glass substrate by a doctor’s blade. The PI-5 coating was obtained by thermally curing the PAA-5 solution in an oven at 80, 150, 250, 300, and 350 °C for 1 h each, respectively. The obtained PI-5 coating was used for thermal and tensile measurements.

Meanwhile, the purified PAA-5 varnish was used to fabricate the quartz CCs according to a procedure reported in the literature [24]. The manufacturing equipment drawing the optical fibers and capillary columns for GCs had a similar procedure. The pristine quartz rod (diameter: 10 mm) was fed into an inert gas-shielded graphite furnace running at a temperature of around 2000 °C. The molten quartz rod was first drawn at a speed of 20 m/min and passed through the PAA varnish tank. Along the draw path, 10 layers of PAA solution were applied in order to achieve a specific thickness. The PAA-coated quartz columns were then thermally dehydrated in a continuous high-temperature oven controlled to be 80–350 °C. The obtained PI-coated quartz CCs were wound with a curvature radius of 10 cm. The thickness of the PI-5 protective coatings was around 25 um.

Other PI coatings and CCs were prepared by a procedure similar to that of PI-5 mentioned above, except that the molar ratio of ODA/4-APBI was 100/0 for PI-1, 80/20 for PI-2, 60/40 for PI-3, and 50/50 for PI-4.

### 2.3. Characterization Methods

Inherent viscosity was measured using an Ubbelohde viscometer with a 0.5 g/dL PAA solution in *N*-methyl-2-pyrrolidone (NMP) at 25 °C. Absolute viscosity was measured using a Brookfield DV-II+ Pro viscometer (Brookfield Ametek, Middleboro, Massachusetts, USA) at 25 °C. Fourier-transform infrared (FT-IR) spectra were measured with a Bruker Tensor 27 FT-IR spectrometer (Ettlingen, Germany) with the scattering wavenumber from 4000 to 400 cm^−1^. For FT-IR measurements, the PAA varnish was coated onto the surface of a KBr plate and cured at elevated temperatures (80–350 °C). Field emission scanning electron microscopy (FE-SEM) was carried out using a Technex Lab Tiny-SEM 1540 (Tokyo, Japan) with an accelerating voltage of 15 KV for imaging. Pt/Pd was spattered on each film in advance of the SEM measurements. Wide-angle X-ray diffraction (XRD) was conducted on a Rigaku D/max-2500 X-ray diffractometer (Tokyo, Japan) with Cu/K-α1 radiation, operated at 40 kV and 200 mA.

Thermogravimetric analysis (TGA) was performed on a TA-Q50 thermal analysis system (TA Instruments, New Castle, Delaware, USA) at a heating rate of 20 °C/min in nitrogen. Dynamic mechanical analysis (DMA) was recorded on a TA-Q800 thermal analysis system (TA Instruments, New Castle, Delaware, USA) at a heating rate of 5 °C /min and a frequency of 1 Hz in nitrogen. Thermomechanical analysis (TMA) was recorded on a TA-Q400 thermal analysis system (TA Instruments, New Castle, Delaware, USA) in nitrogen at a heating rate of 10 °C/min. The tensile properties were tested on an Instron 3365 Tensile Apparatus (Norwood, Massachusetts, USA) with 80 × 10 × 0.05 mm^3^ specimens in accordance with GB/T 1040.3–2006 at a drawing rate of 2.0 mm/min. At least six test specimens were tested for each PI sample and the results were averaged.

## 3. Results and Discussion

Five PI coatings were designed and prepared by the copolymerization of two dianhydride monomers (BTDA and ODPA) and two diamine monomers (ODA and 4-APBI) via a two-step polycondensation procedure, followed by a high-temperature imidization procedure at elevated temperatures, as depicted in Figure 1. PAA varnishes with moderate to high inherent viscosities in the range of 0.92–1.18 dL/g were first prepared, indicating good polymerization reactivity of the monomers (Table 1). PI coatings were obtained by thermally baking the corresponding PAA varnishes cast onto clean borosilicate glass substrates in nitrogen. The PI coatings exhibited good flexibility and toughness with the tensile strength (*T*_S_) of 109.3–137.4 MPa, tensile modulus (*T*_M_) of 2.9–4.7 GPa, and elongations at break (*E*_b_) of 5.0–14.8% (Table 1). It can be clearly observed that the tensile strength and elongation at break of the PI coatings increased first and then decreased when the molar ratio of 4-APBI increased from 0% to 60% in the diamine components. When the molar ratio of APBI was 50% (PI-4), the tensile strength and elongation at break of the PI coatings reached the maximum (*T*_s_ = 141.2 MPa, *E*_b_ = 16.5%). When the molar ratio of APBI exceeded 50% (PI-5), these two parameters began to decrease. The deterioration of the tensile properties of the PI coatings with higher 4-APBI loading is mainly attributed to the relatively higher molecular chain rigidity and lower reactivity of 4-APBI compared with ODA. Nevertheless, the current tensile properties for all of the PI coatings could guarantee their practical applications in quartz CC protection. 

The chemical structure of the PI coatings was confirmed by the FT-IR measurements and the spectra are shown in Figure 2. In the spectra, the characteristic absorptions of the imide rings at 1777 (υ_as__,C=O_), 1722 (υ_as,C=O_), 1376 (υ_as,C-N_), and 721 cm^−1^ (C=O bending) were clearly observed, respectively. In addition, the peaks around 3355 cm^−1^ for the characteristic absorption of –NH– bonds in benzimidazole rings were recorded for PI-2–PI-5, indicating the successful incorporation of the benzimidazole units in the polymers. Meanwhile, the characteristic absorption of the amide carbonyl in PAA at 1640 cm^–1^ disappeared in the spectra, indicating the successful conversion from PAA to PI via high-temperature imidization reaction.

XRD measurements indicated the amorphous structure of the PI coatings. Figure 3 illustrates the wide-angle XRD patterns of the PI coatings, in which blunt absorption peaks were recorded around 2θ = 15°. Incorporation of benzimidazole units did not affect the amorphous nature of the PI coatings.

The appearance of the PI-coated CC samples and the pristine quartz rod is shown in Figure 4. It can be seen that PI coating exhibited good flexibility and toughness, which made it possible to wind the brittle quartz CCs with a curvature radius of 10 cm or even below. The microscopic morphology of the PI-protected quartz CCs was investigated by FE-SEM measurements and the typical photographs of the PI-5-protected sample are illustrated in Figure 5. The average inner diameter of the quartz CCs was 250 um and the thickness of the PI protective coatings was around 25 um. The PI coating adhered with the inner quartz column tightly, and no apparent delamination and cracking occurred. These micrographs also showed that the fracture profile of the PI-coated columns exhibited a typical ductile fracture pattern, indicating good flexibility and toughness of the coatings. At last, the good adhesion of the PI coatings with the quartz CCs was also related to the matched CTEs of the coating with the substrate, which is discussed below.

The thermal and dimensional stability of the PI coatings were investigated by TGA, DMA, and TMA measurements and the data are presented in Table 2. The TGA curves of the PI coating in air are shown in Figure 6. All the PI coatings exhibited good thermal stability up to 500 °C when heating in air. After 550 °C, the PI coatings decomposed rapidly and left 5–35% of their original weights at 700 °C. The thermal stability of the PI coatings was ascertained by the comparison of the 5% and 10% weight loss temperatures (*T*_5%_ and *T*_10%_, respectively). The *T*_5%_ and *T*_10%_ values of the PI coatings increased with the order of PI-1 < PI-2 < PI-3 < PI-4 < PI-5. The thermal decomposition temperatures of the PI coatings increased markedly with the increasing benzimidazole contents. For example, PI-5 had the highest content of benzimidazole units and showed the highest *T*_5%_ value of 553.0 °C in air, which was nearly 11 °C higher than that of the PI-1. Thus, the goal of enhancing the thermal stability of the PI coatings by introducing rigid-rod benzimidazole was successfully achieved. 

The good thermal stability of the current PI coatings can also be reflected by the modulus change during DMA measurements. Figure 7a,b show the storage modulus and loss modulus of the PI coatings at different temperatures, respectively. The PI-1 coating, without benzimidazole component, showed the lowest storage and loss modulus at elevated temperatures, while the PI-2–PI-5 coatings, which incorporated benzimidazole units, had a higher modulus. Further, the modulus of the PI coating films increased with increasing the 4-APBI content. For example, the storage modulus of the PI coatings at 320 °C increased in the order of PI-1(8.1 MPa) < PI-2 (27.9 MPa) < PI-3 (214.6 MPa) < PI-4 (297.7 MPa) < PI-5 (1360.5 MPa). Apparently, the increase of the modulus was mainly due to the incorporation of rigid-rod 4-APBI. From Figure 7b, it can also be observed that the glass transition behavior of the PI coatings could be prolonged by incorporating the benzimidazole units into the PIs. As we know, the glass transition temperature (*T*_g_) is a key factor in deciding if a polymer can meet the demands of specific applications at elevated temperatures. *T*_g_ can usually be defined as the temperature measured by various thermal analysis techniques, such as the peak on loss modulus or loss factor (tan delta) curves in DMA measurements, or the inflection point temperature in TMA measurements. PI-5 showed a *T*_g_ of 335.6 °C (Figure 7b), which was 65.2 °C higher than that of PI-1 (270.4 °C). Thus, the incorporation of rigid-rod benzimidazole units greatly improved the thermal stability of the PI coating. This property will enhance reliability for PI coating applications at elevated temperatures. Figure 7c shows the loss factor (tan delta) plots of the PI coatings, which reveal a similar trend for the glass transition behaviors of the PI coatings at elevated temperatures. PI-1 showed the lowest *T*_g_ value of 278.8 °C and PI-5 exhibited the highest one of 346.9 °C.

Figure 8 shows the TMA plots of the PI coatings. The CTE and *T*_g_ values revealed by the TMA measurements are tabulated in Table 2. All of the PI coating samples were elongated slowly with the increasing temperature until the glass transitions occurred, after which the coatings were elongated rapidly. As expected, a tendency for the *T*_g_ to increase as the 4-APBI content increased was observed, which was consistent with the trend measured by DMA. The CTE values of the PI coatings in the range of 50–300 °C decreased from the value of 62.3 × 10^−6^/K (PI-1) to 24.6 × 10^−6^/K (PI-5). This indicates that the introduction of rigid-rod benzimidazole units into the PI coatings greatly increased the dimensional stability of the polymers. This is quite important for practical applications of PI coatings. The highest *T*_g_ value of 340.1 °C was achieved by PI-5, indicating that it could be used for the protection of quartz CCs operating at the high temperature of 320 °C (20 °C below *T*_g_) for a long time.

The PI-5-coated quartz capillaries were subjected to a thermal impact of 25–320 °C for a total of 100 cycles, which was maintained at each end temperature point for 30 min. After the experiment, the SEM test showed that the adhesion between PI coating and capillary was good, and there was no apparent adhesion failure (Figure 9). The good adhesion of the PI coating with the quartz substrate is mainly attributed to the high dimensional stability of the coating at elevated temperatures and also the strong interactions between the benzimidazole units in the polymer and the surface of the quartz CCs.

## 4. Conclusions

PI coatings with enhanced thermal stability and adhesion for the protection of quartz CCs in GC applications were successfully developed by incorporation of rigid-rod benzimidazole units. PI-5, containing the highest benzimidazole content, exhibited the best comprehensive properties, including the highest *T*_g_ value (*T*_g_: 346.9 °C by DMA and 340.1 °C by TMA), good dimensional stability at elevated temperatures (CTE: 24.6 × 10^−6^/K@50–300 °C), and acceptable tensile properties (*T*_s_: 137.4 MPa; *T*_M_: 4.7 GPa; *E*_b_: 14.8%). These desirable combined properties make the PI-5 coating a good candidate for the protection of quartz CCs for high-temperature GCs operating at 320 °C.

## Figures and Tables

**Figure 1 polymers-11-00946-f001:**
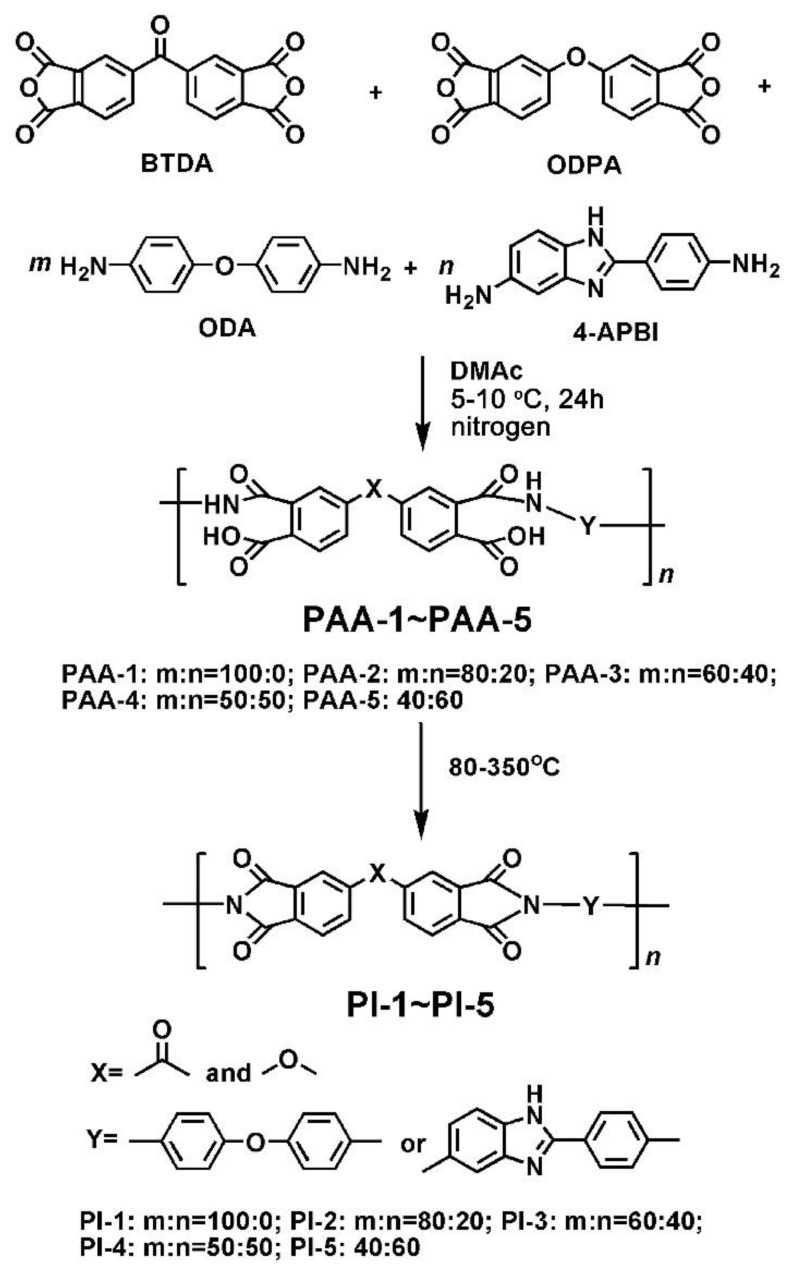
Preparation of PI coatings.

**Figure 2 polymers-11-00946-f002:**
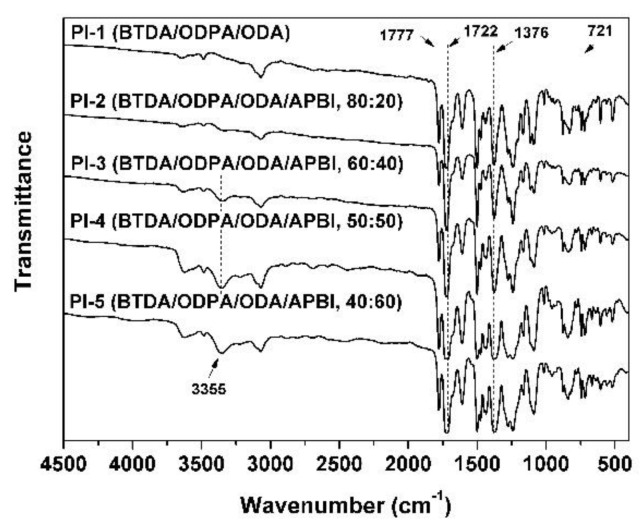
FT-IR spectra of PI coatings.

**Figure 3 polymers-11-00946-f003:**
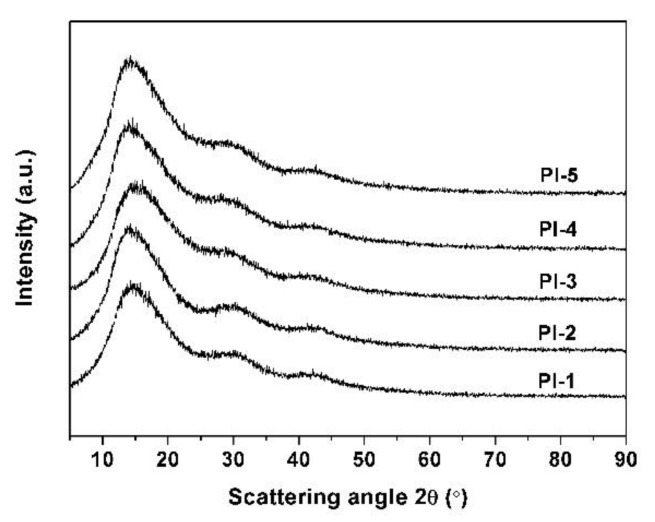
X-ray diffraction (XRD) spectra of PI coatings.

**Figure 4 polymers-11-00946-f004:**
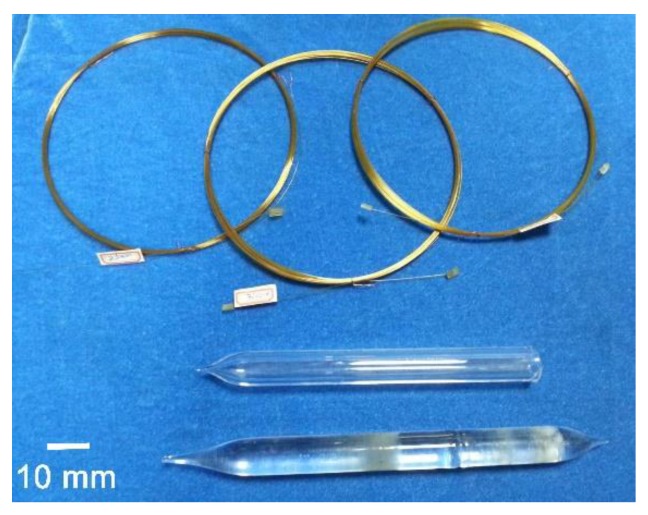
Appearance of PI-coated chromatographic column (CC) samples (**top**) and pristine quartz rods (**bottom**).

**Figure 5 polymers-11-00946-f005:**
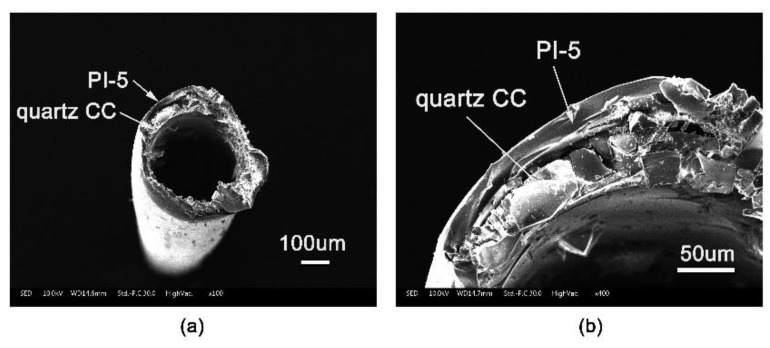
Scanning electron microscope (SEM) images for PI-5-coated CC samples with different magnification times: (**a**) 100 μm; (**b**) 50 μm.

**Figure 6 polymers-11-00946-f006:**
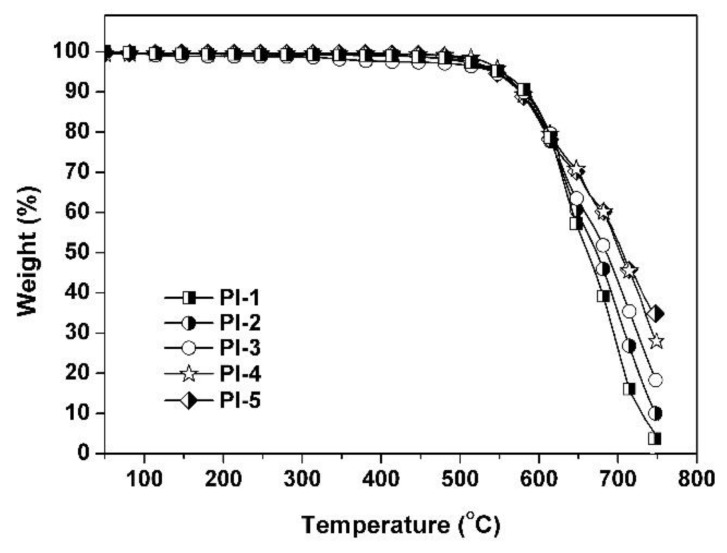
Thermogravimetric analysis (TGA) plots of PI coatings in air.

**Figure 7 polymers-11-00946-f007:**
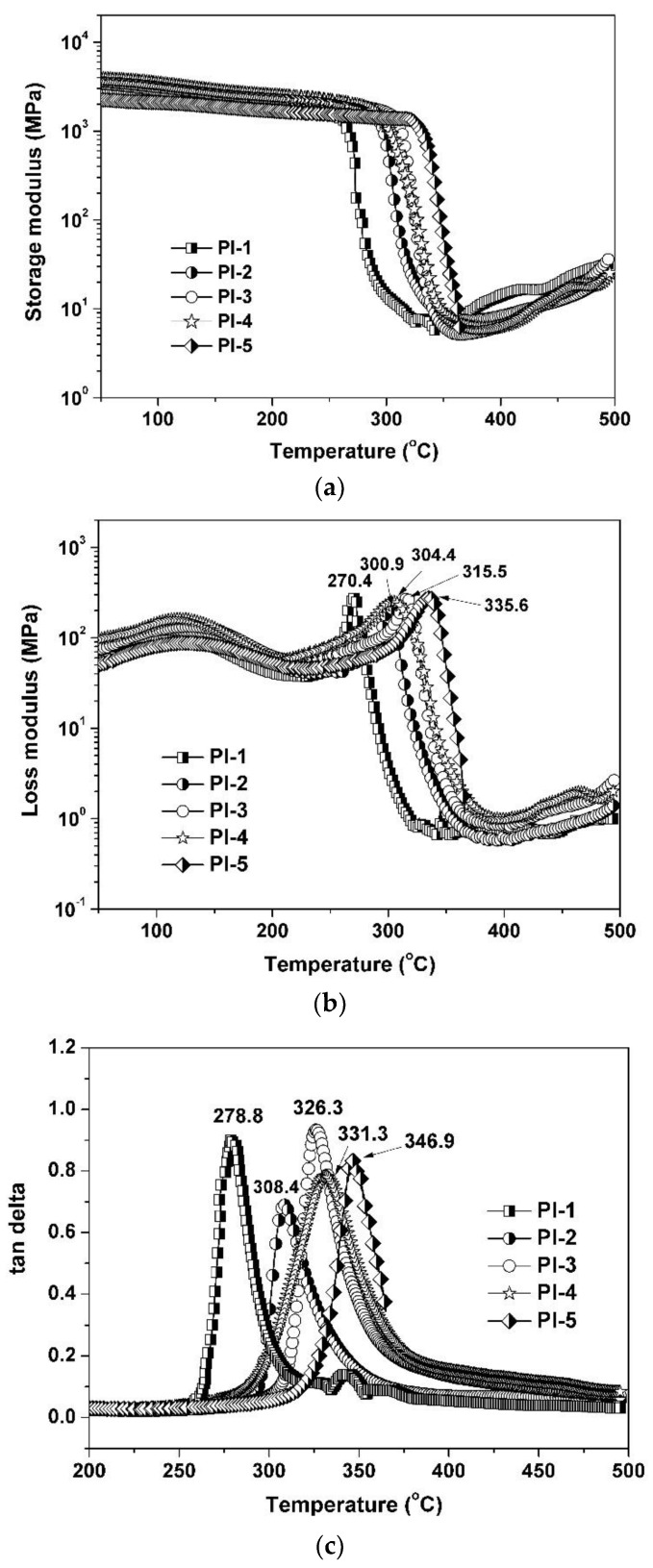
DMA plots of PI coatings: (**a**) storage modulus, (**b**) loss modulus, and (**c**) tan delta.

**Figure 8 polymers-11-00946-f008:**
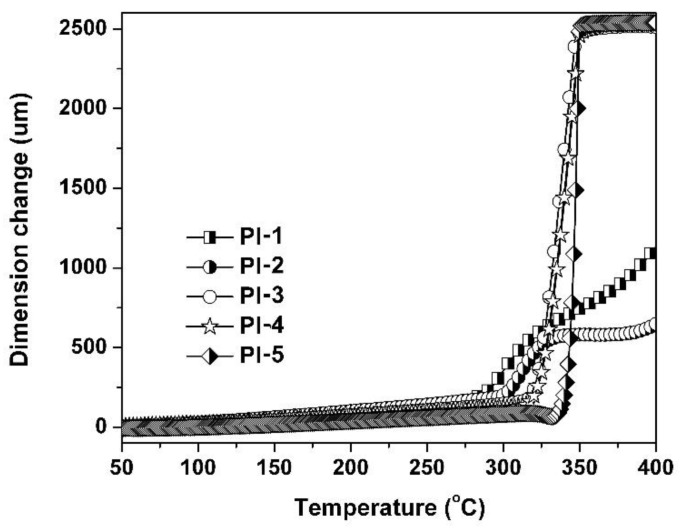
TMA plots of PI coatings.

**Figure 9 polymers-11-00946-f009:**
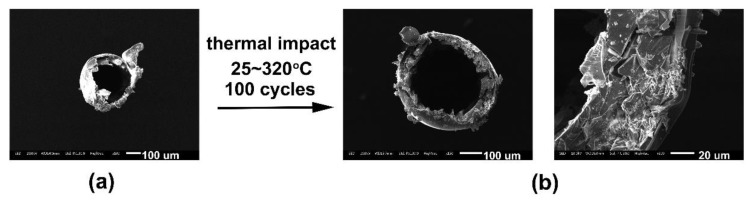
SEM images of PI-5-coated CC samples before and after thermal impact: (**a**) before test and (**b**) after test.

**Table 1 polymers-11-00946-t001:** Inherent viscosities of poly(amic acid) (PAA) solutions and mechanical properties of the polyimide (PI) coatings.

PAA	[*ƞ*]_inh_ ^a^ (dL/g)	PI	Film Quality	*T*_S_^c^ (MPa)	*T*_M_^c^ (GPa)	*E*_b_^c^ (%)
PAA-1	1.18	PI-1	F&T ^b^	109.3	2.9	5.0
PAA-2	1.06	PI-2	F&T	121.0	3.5	9.6
PAA-3	1.02	PI-3	F&T	133.9	4.0	16.5
PAA-4	0.99	PI-4	F&T	141.2	4.4	16.9
PAA-5	0.92	PI-5	F&T	137.4	4.7	14.8

^a^ Inherent viscosities measured with a PAA at a concentration of 0.5 g/dL in *N*-methyl-2-pyrrolidone (NMP) at 25 °C. ^b^ Flexible and tough. ^c^
*T*_S_: tensile strength; *T*_M_: tensile modulus; *E*_b_: elongation at break.

**Table 2 polymers-11-00946-t002:** Thermal properties of PI coatings.

PI	*T*_g_^a^ (°C)	*T*_g_^b^ (°C)	*T*_5%_^c^ (°C)	*T*_10%_^d^ (°C)	CTE (10^−6^/K) ^e^ (50–300 °C)
PI-1	278.8	278.3	542.1	576.2	62.3
PI-2	308.4	299.2	542.6	574.9	58.4
PI-3	326.3	319.6	551.5	577.0	40.3
PI-4	331.3	322.6	552.4	579.3	31.3
PI-5	346.9	340.1	553.0	584.9	24.6

^a^ Glass transition temperature according to the peak temperatures of tan delta in dynamic mechanical analysis (DMA) measurements. ^b^ Glass transition temperature according to the peak temperatures of thermomechanical analysis (TMA) measurements. ^c^ Temperatures at 5% weight loss. ^d^ Temperatures at 10% weight loss. ^e^ Linear coefficient of thermal expansion.

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
