# Peer review of "Preparation and Thermal Evaluation of Novel Polyimide Protective Coatings for Quartz Capillary Chromatographic Columns Operated over 320 °C for High-Temperature Gas Chromatography Analysis"

_polymers, 2019, doi:10.3390/polym11060946_

Round 1
Reviewer 1 Report
The manuscript under consideration deals with preparation and thermal evaluation of polyimide protective coatings for quartz capillary chromatographic columns operated over 320 ℃. It is interesting contribution and worth to be publish after the revision where the authors should discuss also the aging of the coatings described. The aging behavior discussion would be needed prior publication.
Author Response
With respect to the aging behavior of the polyimide coatings, we added the discussion in our revised manuscript as follows. “The good adhesion of the PI coating with the quartz substrate is mainly attributed to the high dimensional stability of the coating at elevated temperatures and also the strong interactions between the benzimidazole units in the polymer and the surface of the quartz CCs.”.
Reviewer 2 Report
The paper by Zhang et al. describes polyimide materials suitable for protective coatings of quartz materials operating at elevated temperatures. The manuscript is well written and its conclusions are satisfactorily supported by experiments. There are only minor typographic remarks.
1) Line 183: Superscript is missing in cm-1.
2) The article graphics were probably originally suggested to fill one column in two-column page. They should be processed to fill better the space in one-column journal. Alternatively, two of them could be merged and denoted a) and b).
Author Response
With respect to the missing superscript in cm-1 in line 183, we modified them in our revised manuscript. With respect to the graphical abstract, we modified it in our revised manuscript in order to show clearly the application of the developed PI coating. According to the instructions for authors of the journal, the graphical abstract was prepared with the size of 11 cm x 9 cm.